# Patient and Therapist Perspectives on Treatment for Adults with PTSD from Childhood Trauma

**DOI:** 10.3390/jcm10050954

**Published:** 2021-03-01

**Authors:** Katrina L. Boterhoven de Haan, Christopher W. Lee, Helen Correia, Simone Menninga, Eva Fassbinder, Sandra Köehne, Arnoud Arntz

**Affiliations:** 1Faculty of Health and Medical Sciences, University of Western Australia, 35 Stirling Highway, Crawley, WA 6009, Australia; chris.lee@uwa.edu.au; 2College of Science, Health, Engineering and Education, Murdoch University, 90 South Street, Murdoch, WA 6150, Australia; Helen.Correia@murdoch.edu.au; 3Department Psychotrauma, PsyQ Beverwijk, Leeghwaterweg 1a, 1951 NA Velsen-Noord, The Netherlands; S.Menninga@psyq.nl; 4Department of Psychiatry and Psychotherapy, University of Lübeck, Ratzeburger Allee 160, 23538 Lübeck, Germany; Eva.Fassbinder@uksh.de (E.F.); Sandra.Koehne@uksh.de (S.K.); 5Department of Clinical Psychology, University of Amsterdam, PO Box 15933, 1018 WS Amsterdam, The Netherlands; A.R.Arntz@uva.nl

**Keywords:** post-traumatic stress disorder (PTSD), trauma-focused treatment, childhood, patients, therapists

## Abstract

This study aimed to explore patients’ and therapists’ experiences with trauma-focused treatments in patients with posttraumatic stress disorder from childhood trauma (Ch-PTSD). Semi-structured interviews were conducted with patients (*n* = 44) and therapists (*n* = 16) from an international multicentre randomised clinical trial comparing two trauma-focused treatments (IREM), imagery rescripting and eye movement and desensitisation (EMDR). Thematic analysis was used to identify key themes within the data. Patients and therapists commented about the process of therapy. The themes that emerged from these comments included the importance of the patients’ willingness to engage and commit to the treatment process; the importance and difficulty of the trauma work, observations of how the trauma focused therapy produced changes in insight, and sense of self and empowerment for the future. In addition, therapists made suggestions for optimising the therapist role in the trauma-focused treatment. This included the importance of having confidence in their own ability, confronting their own and their client’s avoidance and the necessity and difficulties of adhering to the treatment protocols. These reported experiences add further support to the idea that trauma-focused treatments, without a stabilisation phase, can be tolerated and deepens our understanding of how to make this palatable for individuals with Ch-PTSD.

## 1. Introduction:

### Patient and Therapist Perspectives on Treatment for Adults with PTSD from Childhood Trauma

The complex psychopathology of adults with post-traumatic stress disorder from childhood experiences (Ch-PTSD) has been considered as detrimental to the effectiveness of treatment [1]. To date, predominant research focus has been on trauma-focused cognitive behaviour therapies, particularly prolonged exposure approaches, or phase-based treatments [2,3]. Phase-based treatments offer an initial phase focusing on improving patient stability through teaching affect regulation and coping skills prior to trauma processing; however, they have been criticised for delaying individuals from receiving the appropriate treatment for their trauma [4,5]. In addition, despite the evidence base for the effectiveness of trauma-focused treatments, we know that these treatments are not being utilised regularly in everyday clinical practice [6,7]. Given the evident clinical translation gap, there is a clear need to explore different trauma-focused treatments, and to understand what factors facilitate or impede their implementation. Qualitative research methods are well-positioned in this context to help gain a deeper understanding into key processes and barriers to treatments and provide valuable insights for improving their utilisation and effectiveness [8,9].

The most pertinent factors that reportedly interfere with treatment efficacy relate to patients and therapists. Commonly cited contraindicators for patients include concern of symptom exacerbation, avoidance or unwillingness for treatment, and limited distress tolerance skills. It is worth noting, however, that these patient-related treatment issues are largely based on the opinions of therapists [10,11,12]. Where studies have focused on patient treatment preferences, patients tend to opt for trauma-focused strategies over other psychotherapies, and the opportunity to talk about their trauma [13,14]. In contrast, therapist-focused research has reported therapists’ own reluctance to implement trauma-focused treatments with concerns including fear of “opening Pandora’s box”, limited belief in the credibility of treatments, or doubting their own competence [15,16,17].

The aforementioned research has so far mainly focused on prolonged exposure approaches and therefore the findings may not be generalised to other trauma-focused approaches. Two alternate treatments are imagery rescripting (ImRs) and eye movement desensitisation and reprocessing (EMDR). EMDR uses bilateral stimulation such as therapist hand movements or tapping to reconsolidate trauma memories and help resolve emotional distress [18]. In ImRs, patients are guided to rescript their trauma memory with a more acceptable ending that helps to meet the patients’ needs, which leads to fundamental changes in belief systems and serves to alter the meaning of the trauma [19]. ImRs and EMDR are trauma-focused treatments which limit exposure to trauma memories by only initially focusing on the trauma experience before processing, thus potentially making them more preferable to patients and therapists than traditional approaches [20,21].

Given there appears to be reluctance to engage in trauma-focused treatments from both patients and therapists despite their effectiveness, research into exploring both perspectives could prove informative. There are very few studies that have integrated and synthesised patient and therapist views on PTSD treatment, i.e., [22,23]. In terms of adults with PTSD from childhood, only one study was identified [24], however it did not involve a trauma-focused treatment. With this in mind, the purpose of the present study was to explore both patients’ and therapists’ perspectives of the Ch-PTSD treatment experience using two different trauma-focused approaches to elucidate experiences that can improve current practices.

## 2. Method

### 2.1. Study Context and Design

Study participants were patients and therapists involved in an international randomised clinical trial investigating the effectiveness of ImRs and EMDR (IREM) in the treatment of childhood trauma-related PTSD. The trial was registered with the Australian New Zealand Clinical Trials Registry (ACTRN12614000750684) and ethics approval for the study was obtained from the relevant ethics committee in each country with protocol numbers Government of Western Australia, department of health (2014067EW), Maastricht University (ECP-136_01_01_2014), and University of Lubeck (14-274).

In IREM, patients were randomly allocated to either ImRs or EMDR treatment condition which included 12, 90 min sessions, twice weekly for a period of six to eight weeks. Eligible participants had a primary diagnosis of PTSD related to trauma experienced before 16 years of age. Participants were screened for inclusion and exclusion with the use of the Structured Clinical Interview for DSM-TR [25] or Mini Neuropsychiatric Interview [26], depending on site preference. Exclusion criteria included benzodiazepine use, acute suicide risk, comorbid psychotic disorder, bipolar disorder type 1, alcohol or drug dependence, and no trauma-focused treatment within the past 3 months.

Therapists provided either ImRs, EMDR, or both treatments to IREM patients. The treatments followed written treatment protocols. Each therapist was required to attend a 2-day advanced workshop in the chosen treatment following basic training and attend regular supervision throughout the trial. Additionally, therapists were required to demonstrate competence in the treatment prior to treating IREM participants by seeing pilot patients, which was video recorded and assessed by the local site coordinator. Each therapist also had regular supervision by peers and were encouraged to use fidelity checklists to ensure they were adhering to the treatment protocols. Full methodology and design for IREM is outlined in an earlier publication [27]. The IREM treatment outcomes are reported in [28].

### 2.2. Participants

A sub-sample of IREM patients and therapists was used for this qualitative study. Overall there were 44 patients who accepted the invitation to be interviewed and 16 therapists who treated patients as part of the IREM RCT. Demographic information for participants used in this study are displayed in Table 1 and therapist characteristics are displayed in Table 2. Therapist experience in therapy overall ranged between 5 to 36 years and experience working with Ch-PTSD ranged between 0 to 28 years.

### 2.3. Semi-Structured Interview

A semi-structured interview addressing the study aim was developed by author KBdH, informed by the literature, and reviewed by authors HC and CL. Interview questions were revised iteratively using feedback from multiple rounds of pilot interviews with patients and therapists who had experience of the treatments, but were not involved in the IREM analysis. Interview questions related to overall experiences, nature of change, the treatments, and childhood trauma treatment. Examples of interview questions include “What do you think are the important components of therapy that facilitate change” and “Can you tell me about the therapy you received and your thoughts about it?” The list of questions are provided in the Appendix A.

### 2.4. Research Assistants

Research assistants were post-graduate level students or PhD candidates. Each research assistant was given training by authors KBdH or SM on the aims of the study, how questions should be asked, and the use of prompts. Research assistants conducted an initial interview, which was then transcribed and reviewed by KBdH and SM who provided feedback. Both KBdH and SM were therapists involved and treated patients as part of the IREM trial. They did not interview any patients that they directly treated, however they did conduct interviews on patients treated by other study therapists.

### 2.5. Procedure

The results of this study are reported according to COnsolidated criteria for REporting Qualitative research (COREQ) guidelines [29]. Patients from four different sites (Australia, Germany and two sites in the Netherlands) were approached to participate in the interviews following the 8-week post-treatment assessment. All patients receiving treatment between October 2016 and June 2018 were contacted and asked if they were willing to participate and those that agreed were then scheduled for an interview. Therapists from each site were initially emailed by their site coordinator and a follow-up email by author KBdH to request their participation. Participant information sheets were provided to every person who agreed to be interviewed and signed informed consent was obtained.

Therapist interviews were all conducted by author KBdH in English. Patient interviews were conducted in the language of the site and were carried out by authors KBdH, SM, and trained research assistants. Interviews were audio-recorded and lasted approximately 60 min. At the conclusion of each interview, a verbal summary was given by the interviewer where feedback was encouraged. After interviews were transcribed, a checking process was utilised where a written summary of interview transcripts was provided to participants asking for corrections or feedback. No patients made any changes to interview transcriptions as a result of the checking process. Where applicable, interviews were then translated to English and the finalised transcripts and interview summaries were uploaded to MAXQDA software 2018 (VERBI Software, 2017) [30].

### 2.6. Data Analysis

While the patient and therapist groups were treated separately, the data analysis process followed the same methodology. For the purposes of the current study, data across treatment conditions was combined as the focus was on elucidating elements of trauma-focused treatment for Ch-PTSD to enhance treatment for this population. The research team used both an a priori directed approach, which was influenced by topic areas of the interview questions, and inductive identification of codes and themes across topics using thematic analysis as it allowed for open, flexible coding to identify themes relevant across different groups and sites [31]. To enhance credibility and confirmability in the development of codes and themes during the analysis, the multi-step analysis process was an iterative series of individual coding, peer debriefing, researcher triangulation, and testing for referential adequacy [32]. To reduce potential bias, both the research assistant involved in the coding and author HC were independent of the IREM study.

Author KBdH and a research assistant, became familiar with the data, reading and re-reading transcripts to identify, analyse, and record patterns [31]. Independent line-by-line analysis was used to identify preliminary codes from an initial pool of interviews. These interviews were also analysed by author HC with peer debriefing around initial impressions of codes and a coding framework. The stages of identifying emerging themes and reviewing themes followed a similar iterative process of reviewing and consulting with authors HC and CL, testing for referential adequacy by returning to the raw data [32]. These stages of analysis focused on identifying underlying codes and themes across interview topics. Themes were identified at a semantic level (realist method) where analysis focused on describing and interpreting the explicit or surface meanings of the data [33]. Themes were then more clearly defined, identifying relationships between themes and subthemes.

Evidence-based recommendations for topic saturation suggest that saturation occurs within the first 12 interviews, and that basic meta-themes are identified as early as six interviews. Furthermore, it was expected that each group would have a relatively high level of homogeneity, whereby topic saturation would be achieved sooner than 12 interviews [34].

## 3. Results

Analysis of the interviews resulted in the identification of super-ordinate and subordinate themes related to the treatment process and for optimising the role of the therapist (Figure 1). Patients and therapists had many shared views in relation to the treatment process including: preparation for treatment, the importance of treating the trauma, and the nature of change. The super-ordinate theme of optimising the therapist role was unique to therapists, who identified elements which can impact treatment including the therapeutic relationship, their confidence, confronting avoidance, and adherence to the treatment protocol.

### 3.1. Focusing on Trauma Memories

Focusing on trauma memories was considered important to both patients and therapists in this study. This super-ordinate theme is related to trauma-focused treatment to address patients’ childhood experiences. Aspects of this theme included the need to go back to the original trauma experience, the difficulty in going back to the source, and having a corrective emotional experience.

#### 3.1.1. Willingness

Both patients and therapists discussed the need for willingness. This willingness was related to the readiness of the patient to commit to treatment, and to addressing their trauma:


*Participant 19: “Of course the willpower that you really want to change something. That is the most important. If there is no willpower, it won’t work. If I want to change something and work on such a trauma, then I really have to want it.”*



*Participant 9: “I think this treatment would work the vast majority of cases, but… everybody is different... I got out of it, because I gave a lot into it. I came here when I didn’t want to, and would still jump in … and still get involved, as how hard it was.”*



*Therapist 16: “Well, certainly the motivation... the readiness, you know, to do trauma work. It’s helpful, you know, just in terms of people’s ability to come and commit and to tolerate the work.”*


#### 3.1.2. Starting Trauma Work

Therapists and patients discussed when was the right time to start doing trauma work. IREM required trauma processing from session two onwards. Patients and therapists had mixed views. For example, a few therapists thought it would have been better to have a bit more time to build a relationship with the patient or to have some time for skill building with patients:


*Therapist 11: “I think in a lot of cases, it’s best to start as soon as possible with trauma-focused therapy. But I guess in some cases, it’s better to first build some skills and use the phase-based therapy. Then people are able to cope with the possible effects of the trauma therapy.”*


However, most participants though that it was better to start trauma work sooner and did not see the need for prolonged preparation:


*Participant 23: “I knew what I was starting. Of course you never know how it’s going to work. But you know what it’s for and what exactly is going to happen. I thought the explanation, the information you get… I don’t think more is necessary because then you’re going to think about everything.”*



*Participant 30: “You know, in my case, if it (the introduction) were (any) longer… then I would get scared or confused. I would’ve said I don’t want to do it anymore.”*



*Therapist 2: “I’m now having had this experience of the research I’m quite surprised that we could go in there and do stuff so quickly and have good results without poor outcomes with actually good outcomes and now and then occasional situations there were risks that—were actually managed.”*


#### 3.1.3. Going Back to the Source

For patients, they felt it was important that treatment went back to the original trauma experience. Patients and therapists both recognised that the original event was related to the onset of the patients’ current difficulties and this needed to be reprocessed to help change their perspective and beliefs around the experience and to help them move forward:


*Participant 27: “I don’t know how you would treat it otherwise. It is just something that had a big impact on your life. Something that you weren’t able to deal with correctly. So the only way to fix that is to go back and do it over, I think.”*


Similarly, most therapists thought that is was essential for patients to be exposed to memories and feelings associated with their trauma, and then having a corrective emotional experience where they learn to view themselves and their traumatic past in a different way:


*Therapist 15: “What (is) helpful is that... they go into the trauma again and have a new experience in their brains and then they start to think in a different way. They don’t feel guilt anymore because it’s not a secret anymore.”*


While patients were able to recognise that going back to the trauma was an important component of treatment, most of them acknowledged how difficult this was for them and others commented that whilst this experience was difficult it was very important part of the treatment:


*Participant 10: “It was difficult but I feel that it needed to be done that way... I think, it was kind of like kind of a slap in the face but a good one. So, what I am trying to say is, yes, it was hard going back like that … but I think that was partly necessary.”*


Some therapists also discussed the difficulty they experienced during trauma processing.


*Therapist 5: “Not to fear your emotions and you have to be able to stand for minutes, and minutes, and minutes and see that somebody is not going well, not at all and to accept that, you do not have to do the changes. It’s not your job to change things, it’s the patient’s job to try.”*


#### 3.1.4. Enhancing Treatment Format

The IREM treatment format involved 12, 90 min sessions, twice a week for a period of up to eight weeks. Both patients and therapists were positive about the actual treatments; however, they did offer some insights and suggestions for improving the overall treatment format. Therapists and patients considered the number and frequency of treatment sessions.

For some of the patients, they felt that 12 sessions were not enough for them:


*Participant 8: “I think if there was a two or four more to just see me through, just to help me just get through the last few doors you know, because I still struggle with things.”*



*Therapist 4: “I often have the feeling in the 12 sessions they need so much more.”*


However, not all agreed that more than 12 sessions were needed.


*Therapist 13: “I’m not sure. I thought it was okay. With the patient I treated, it was really okay. We almost didn’t know what to address at the end. We almost had sessions open. With the other patient, … She was in process but it was just she needed more.”*


While no patients specifically commented that they felt they needed less sessions, in the entire sample of patients treated in IREM (n = 155), there were 18 participants who finished treatment early.

Session frequency was another issue discussed in the interviews. Some of the patients and therapists preferred the twice weekly session approach used in IREM. Patients and therapists discussed how the intensive approach helped promote patient engagement and it facilitated processing:


*Participant 31: “I didn’t experience anything negative about it. It’s hard, twice a week, but not uhm… it’s very effective, a lot has happened.”*



*Therapist 7: “I think it’s good. I think it’s good because yeah, it is, everything is so fresh in the patient’s memory. You can easily come back to what you have done in the session before … And I think it’s good for patients to have a short intense of treatment. I think it’s more effective than just feeling it really. To me it feels more effective than 12 sessions and 12 weeks. Yeah, but at the same time I found it pretty intense for me too and its very time consuming.”*


However, not all agreed that twice weekly sessions were better, with comments that the intensity of the treatment was too much for some:


*Therapist 15: “Sometimes it’s too busy for us but I think it’s very good that they come twice a week but there are sometimes for whom it would be better to come once a week I think. That are clients who have emotionally not that much control and they need few days’ rest.”*


### 3.2. Nature of Change

The change process involved patients’ giving context to their trauma, which was through improving their understanding of the circumstances and their role in the events, and helping them to develop a new perspective about their experiences and themselves. This process led to greater self-acceptance, hope for the future, and patients feeling strong and in control of their life.

#### 3.2.1. Trauma in Context

Therapists suggested that patient change was about them gaining a new understanding of their experiences and how this has impacted them. It was felt that gaining a new perspective would help to change their beliefs about themselves and the trauma, which would lead to them developing a sense of agency over their lives and behavioural change:


*Therapist 9: “Well, I do think that they very often recognise what they missed and it’s however painful … So instead of thinking, they were the dirty child or the guilty child, they see that they were a child in very bad circumstances and that of course when you face that and you really feel something about it then afterwards you get more mild towards yourself.”*


For patients gaining a new perspective or insight was about them learning that what happened was not their fault and they were not to blame:


*Participant 21: “How I look at myself in those situations has definitely changed… I can now honestly say that I did what I could as a child... there was nothing I should have done to begin with, so I didn’t fail to do anything. Or neglect to do anything. That I’m less hard on myself, in that sense. I am less critical of myself.”*


By developing an understanding of their role in their trauma it facilitates a changes in their relationship with the trauma or how patients view themselves:


*Participant 16: “I do think differently about myself. I don’t see myself as this monster, Frankenstein, this massively damaged and broken thing. I still see the injuries I sustained but now I know they can heal, at least to a certain degree, and they can even have positive effects. I would surely not be the artist I am today without those experiences... I did see it before therapy but now I believe it.”*



*Participant 11: “I think after the treatment I felt more empowered and like even though I couldn’t do anything I could do something now to help myself. So, it really made me feel like in control of what happened, if that makes sense, in a weird way.”*


#### 3.2.2. The Changed Self

Once patients were able to address their distorted thinking of the trauma, it helped to change their view on themselves and their future. For most patients, a focus on the future was a new experience and gave them a sense of hope that things could be different for them:


*Participant 23: “Let’s just say, it wasn’t in my nature. The possibility of a future … I didn’t see any possibilities. Because there weren’t any possibilities. And now I think differently. The possibilities are different.”*


Most therapists reported that patients gaining a sense of empowerment and autonomy was crucial for therapeutic change to occur. This change related to patients’ drawing on their inner strength and resilience by taking control of themselves and their future:


*Therapist 5: “They find a lot that they are able to look at that they are strong enough... are able to experience the feelings, able to experience the ideas and the pictures and the thoughts and... their role changes and they’re not a victim anymore.”*


Patients discussed how treatment had given them the strength and the ability to change. This led patients to start taking control, asserting themselves, and embracing a life that they had never thought possible:


*Participant 25: “I am stronger, I think more clearly, I’m not afraid anymore … I can enjoy things more, I have a less internal battles … almost none, and if there is one then I can handle it well.”*



*Participant 19: “I don’t put up with everything … I just say: ‘hey, I’m a human being as well with feelings and I’m valuable’ and I don’t have to be there for others all the time... A lot of times I said ‘the most important thing is that the others feel fine, I don’t matter’ but now...’ I won’t do that. I don’t want to do that.”*


### 3.3. Optimising the Therapist Role

Optimising the therapist role contained subordinate themes only discussed by the therapists in the study and were factors specific to trauma-focused treatment. Sub-ordinate themes included therapist confidence, avoidance, and adherence.

#### 3.3.1. Therapeutic Relationship

Therapists identified the therapeutic relationship as important to help create a safe and supportive environment so patients felt able to engage in trauma-focused treatment. Therapists reported that this relationship helped to facilitate change:


*Therapist 10: “It’s the working alliance... there’s enough safety in the structure and the rules. It’s clear what’s going to happen, that the patient has control or is in control.”*



*Therapist 15: “If you have good contact and… they trust you, … people will change because in time they trust themselves and feel comfortable… then they can change by themselves, I don’t change.”*


The importance of the therapeutic relationship was not shared by all therapists:


*Therapist 4: “I think the relationship between patient and therapist is important. I also think it’s sometimes overrated. It’s important but it’s not for every patient that important but a lot of therapists think it is. Just my opinion.”*


Only a few of the patients specifically discussed the therapeutic relationship. Many did, however, comment on the characteristics of the treatment or therapist that they found important. Some examples include therapists being empathetic, knowledgeable, and making patients felt safe.

#### 3.3.2. Therapist Confidence

Confidence was a theme which all therapists acknowledged impacted on treatment. Therapists talked about having confidence not only in themselves and their ability but also in their patients’ capacity for treatment:


*Therapist 14: “I think that it’s important that the therapist is not afraid and can give patients a feeling of confidence… like a doctor,... it’s important that the doctor gives you the idea that he knows what he or she knows what he does and that it’s ok and that he is in control.”*


Therapists also identified the importance of having confidence in the ability of the patient to tolerate the treatment:


*Therapist 3: “Don’t be afraid to kind of push your client… I think there is a lot of possibility going on and look I think,... if you give the client that option, they can do it.”*


#### 3.3.3. Avoidance

Avoidance was a theme that arose in all of the therapists’ interviews and was relevant to both patients and therapists. Therapists commented on patient’s avoidance of trauma memories and the feelings associated with them:


*Therapist 11: “I think that is one of the most important part that they are not running away from it and not putting it away in their mind.”*


Some therapists acknowledged their own avoidance of treatment, which was related to fear of symptom exacerbation:


*Therapist 13: “What I really felt in our centre is that therapists are very reluctant to do trauma-focused therapy with these patients that they feel that you need to protect them... They’re afraid to de-stabilise this patient.”*


#### 3.3.4. Adherence

For therapists, avoidance was closely linked to treatment adherence. Adherence to the treatment protocol was found to help to overcome avoidance. Therapists discussed that outside of the trial they would have stopped treatment when patients became too distressed; however, they found that having to adhere to the treatment protocol actually facilitated treatment and reduced patient distress sooner:


*Therapist 16: “The really good thing about it is that it was really structured and it forced us to get going much sooner than we normally would. You had to do processing in every session which has been great because I know that outside of it I get far too easily side tracked and we are distracted by crisis and stuff and avoid it.”*



*Therapist 10: “That we are used to calm down the client or do some relaxation exercise, but the protocol says, well, you should go on… We are used to taking care… (patients) can take time out. But afterward, I see well, it was right… Just to keep going.”*


Some therapists, however, did struggle with adherence and felt that it impacted on their ability to provide treatment.


*Therapist 5: “Well you have to stick to the protocol, but you are not allowed to use them (strategies), what I feel that there’s something missing and I cannot do it, that’s unhelpful for me.”*


## 4. Discussion

In this study, patients and therapists were asked their perspectives on receiving trauma-focused treatment for Ch-PTSD. Analysis of the transcripts led to key themes related to the process of trauma therapy and comments for therapist on how to optimise their behaviour to achieve the best outcome. In commenting on the process of therapy, similar themes from both patient and therapist interviews emerged and included ideas on the importance of preparation, focusing on the source of the trauma, and providing therapy twice a week. Key aspects of the change process were also described with patients gaining a new understanding of their trauma and how this has impacted them, developing their inner strength and gaining a sense of hope for their future, all of which lead to improved self-efficacy and self-worth. Both therapists and patients, whilst commenting on the benefits of directly processing trauma memories, also reported that the process itself was challenging. Therapists identified aspects of their behaviour which they think impacted the treatment. They commented on the need to have confidence in their ability to effectively treat patients, the importance and difficulty of adhering to treatment protocol, and the tendency to want to avoid trauma processing.

In general, these qualitative findings were positive about the benefits of trauma-focused treatment for Ch-PTSD. They are consistent with the observer-rated quantitative findings and self-reported symptom improvement of patients in the larger IREM study [28]. IREM findings identified significant reductions in PTSD symptoms post-treatment with 68% no longer meeting PTSD criteria at the 8-week follow-up assessment. A range of secondary measures also found a significant reduction in patients’ symptoms including self-reported PTSD, depression, dissociation, anger, and trauma-related cognitions and feelings such as shame and guilt. In terms of dose, the 12 sessions were sufficient for most participants. While some might have felt they were not ready to end, they went on to make further gains which was evidenced in the 81% recovery from PTSD at one year follow up.

Preparation was a theme identified by our patients and therapists. The therapists emphasised the therapeutic relationship as being important in preparing patients to engage in treatment by providing a safe environment where they are supported and in control. Patients and therapists both discussed the willingness of the patient for treatment which was more specific to the internal characteristics of the patient. Overcoming patient ambivalence for trauma processing and having a safe and trusting environment during treatment was discussed by Shearing and colleagues [35], who investigated patient views on reliving the trauma as part of their PTSD treatment. Of interest, the results of this study were similar to that of other studies which included treatments with a stabilisation phase or as part of a group program, suggesting that the same treatment components and difficulties exist regardless of the inclusion of a phase-based approach [36,37]. One argument for phase-based treatments is that they help reduce patients’ fears and promote their receptivity to trauma processing [38]. However, while our findings support strategies for preparing patients for treatment, we found that these could be achieved without a separate stabilisation phase in treatment. IREM results indicated that they were able to tolerate any potential fear of trauma processing in that the dropout rate was only 7.7%, suggesting that the treatment was acceptable for patients without any prior stabilisation [28].

Patients and therapists considered that confronting and reprocessing of trauma memories was crucial for change. Patients recognised that trauma processing was difficult but that there was no way to move forward without processing the thoughts, feelings, and experiences of their past. Several studies that have explored treatment preferences have also found that patients are willing to brave the effects of trauma-focused treatment as confronting their memories was needed before they could successfully move on [13,39]. The importance and need for confronting past trauma experiences has been consistently identified across the research investigating patients’ treatment perspectives. This suggests that the benefit of addressing their past experiences far outweighs patients’ fears of reliving their trauma and is encouraging for promoting trauma-focused interventions [35,40].

Similar to other research, our patients and therapists viewed the experience of dealing with their trauma helped to provide a context for understanding feelings and patterns of behaviour; to change patient’s perspective, particularly of themselves; as a result, they felt stronger and more in control of their lives [8,41]. In particular, our patients’ talked about gaining a sense of hope for their future, learning to accept themselves, starting to consider their own needs, and being more assertive. Key themes identified in other studies describe a process of healing and recovery where patients take an active role in their treatment, draw on internal resources, and develop a sense of agency [42,43]. Together, there is a clear recognition that therapists should actively consider how to enhance engagement in way that promotes a sense of agency which is particularly relevant to individuals with Ch-PTSD.

Relatively novel in this study was the exploration of therapists’ perspectives. To deliver trauma focused interventions with this population, therapists identified the importance of being confident in their approach. They also recognised that they can avoid doing the actual treatment, with a number citing that adherence to the treatment protocol helped them to manage the sessions. For example, many therapists acknowledged that in the past they would have been guided by patients’ presentation at each session, including their avoidance, or that they might have stopped treatment when they could see that their patients were particularly distressed. However, having to adhere to the research protocol served therapists in reducing avoidance and in “staying the course” during therapy, thus not changing the way treatment delivery is intended. Although not all therapists found adhering to the protocol as beneficial, in particular some found that it interfered with meeting the needs of the patients. Therapists also acknowledged that gaining confidence in their own ability made them feel more competent in the management of any issues that arose from trauma processing. One of the main aims of this research was to identify issues related to poor implementation and utilisation of recommended trauma-focused treatments. In IREM, the therapists had comprehensive training and peer supervision throughout the trial which may have contributed to increased confidence in their ability, the effectiveness of the treatment, and the capacity of the patient to tolerate treatment. Our findings did not present any significant differences between level of confidence and therapists’ level of experience. Research has shown that novice clinicians can effectively implement PTSD treatments and produce good treatment outcomes [44]. Taken together, it suggests that therapist experience does not impact treatment outcomes but highlights the effect of their level of confidence, avoidance, and adherence on treatment.

IREM incorporated an intensive short-term trauma-focused approach to treatment. Most of the IREM patients and therapists were supportive of this intensive approach, in particular twice weekly sessions. Studies have found that the majority of patient dropouts occur before mid-treatment and hence, intensive treatment approaches potentially increase the likelihood of early symptom improvements, which may contribute to better attrition rates [45]. It is possible that the intensive format contributed to the low IREM dropout rate; however, more research is needed and is currently being investigated (see Netherlands Trial Register NTR7153).

### Study Limitations

Irrespective of the interesting findings arising from this study, it is not without its limitations. In relation to the patients in the IREM trial, they had already been through the screening process and initial assessments including agreeing to 12 sessions of trauma processing and identifying target memories before they began treatment. Therefore, the experiences our patients and therapists had towards the trauma-focused approaches may not generalise to other Ch-PTSD patients. Despite this, it is possible that their level of motivation or readiness for treatment had developed because of the intake process. As such, this process could be implemented in regular practice, to stimulate patients’ commitment and readiness for change.

A possible separate limitation is that apart from the Australian participants, all spoke English as a second language. The patient interviews were conducted in the language of the site and were then translated to English and the therapist interviews were conducted in English. As a result, there may have been language or cultural nuances lost in translation. However, we attempted to address any possible misunderstandings through our patient and therapist checking procedure where each interview was summarized and sent to participants for clarification or comment.

## 5. Conclusions

This is the first study that directly integrates and synthesises patient and therapists’ views of trauma-focused approaches for treating adults with PTSD from childhood trauma experiences. This research makes a valuable contribution to improving our understanding of important components of trauma-focused interventions, the nature of change, and treatment-related issues and possible ways for improving their effectiveness. Patients and therapists from three different countries identified several shared views on components of trauma-focused treatments important in treating Ch-PTSD. Treatment mechanisms identified by both patients and therapists emphasised the importance of addressing trauma experiences so that patients can gain a new perspective into their trauma and how this has impacted them, to develop a sense of agency and have hope for their future. Therapists acknowledged that their confidence can impact the effectiveness of treatment, which can contribute to greater avoidance.

While there is growing evidence that supports the use of trauma-focused treatments for Ch-PTSD, these approaches are under-utilised. Future research would benefit from more investigation into the ways of improving therapists’ confidence and overcoming barriers to treatment, and the impact of intensive trauma interventions to find out how this can effect patients’ treatment engagement and outcomes.

## Figures and Tables

**Figure 1 jcm-10-00954-f001:**
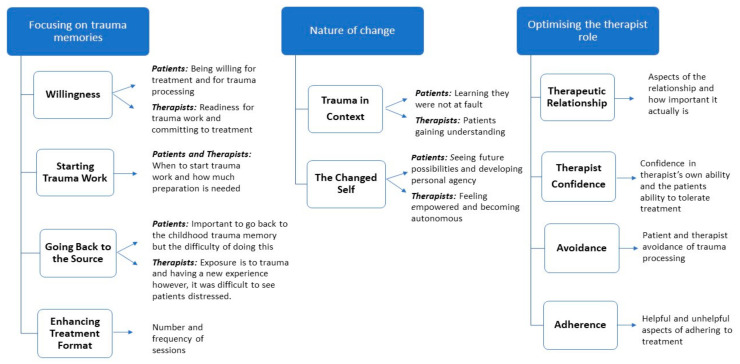
Patient and therapist shared super-ordinate and subordinate themes on the treatment process.

**Table 1 jcm-10-00954-t001:** Demographic information and characteristics of IREM patients.

Characteristics	No. (%) of Patients ᵃ
	*n*= 44
Age, Mean (SD), years	40 (12.16)
Sex	
Female	32 (72.7)
Male	12 (27.3)
Country	
Australia	12 (27.3)
Germany	25 (56.8)
Netherlands	7 (15.9)
Relationship Status	
Partner	28 (63.6)
Education level	
High school or less	13 (29.6)
College or above	31 (70.3)
Work status	
Working	19 (43.1)
Disability pension	11 (25.0)
Unemployed	9 (20.5)
Other	5 (11.4)
Psychological History	
PTSD duration, Mean (SD), months	237.82 (183.54)
Co-morbid mood disorder	32 (72.7)
Co-morbid anxiety disorder	25 (56.8)
Previous Treatment	34 (77.3)
Previous Psychiatric Admission	21 (47.7)
Index Trauma History ᵇ	
Sexual Abuse/Assault	23 (52.3)
Physical Abuse	13 (29.5)
Other	8 (18.1)
Index trauma onset, Mean (SD), years	7.57 (3.78)
Index trauma duration, Mean (SD), years	8.25 (4.50)

ᵃ Percentages have been rounded and may not total 100. ᵇ Index trauma relates to trauma experienced before 16 years of age.SD: standard deviation; PTSD: Posttraumatic Stress Disorder

**Table 2 jcm-10-00954-t002:** Characteristics of IREM therapists.

Characteristics	No. (%) of Therapists ᵃ
	*n* = 16
Age, Mean (SD), y	42.38 (7.84)
Sex	
Female	14 (87.5)
Male	2 (12.5)
Country	
Australia	3 (18.8)
Germany	4 (25.0)
Netherlands	9 (56.3)
Theoretical Orientation	
CBT	15 (93.8)
Schema Therapy	10 (62.5)
EMDR	7 (43.8)
DBT	5 (31.3)
Therapy Experience, Mean (SD), y	
Years of practice (overall)	16 (8.97)
Years with Ch-PTSD	7.13 (6.78)

ᵃ Percentages have been rounded and may not total 100. SD: Standard Deviation; CBT: Cognitive Behaviour Therapy; EMDR: Eye Movement Desensitisation and Reprocessing; DBT: Dialectical Behaviour Therapy; Ch-PTSD: Posttraumatic Stress Disorder from childhood trauma

## Data Availability

The data that support the findings of this study are available on request from the corresponding author. The data are not publicly available due to privacy or ethical restrictions.

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
