# Peer review of "Patient and Therapist Perspectives on Treatment for Adults with PTSD from Childhood Trauma"

_jcm, 2021, doi:10.3390/jcm10050954_

Round 1
Reviewer 1 Report
Unfortunately, the Authors responded to only a few comments from the previous review. Below is a list of comments:
There is still no line numbering and the diagram presented is not clear. Also, Table 2 has become unreadable
The authors also did not answer, on what basis the number of sessions was 12? Why is this? Have any preliminary studies been conducted? Could it be indicated by the literature data?
"KBdH, informed by the literature, and reviewed by authors HC and CL" - Researchers' abbreviations should be introduced into the publication like all other abbreviations, ie they should be expanded on first use.
Are there references to studies performed [28] on studies conducted in the same population? It is not explained where the health information came from (end they go on to make further gains which was evidenced in the 80% recovery from PTSD at one year follow up) since the interview was conducted 8 weeks beyond the end of the session. The authors also did not respond to this comment.
While no patients specifically commented that they felt they needed less sessions, there were 18 participants of the 155 treated who finished treatment early. - why does the number 155 appear?
Author Response
Review Feedback and Response
Reviewer 1
Unfortunately, the Authors responded to only a few comments from the previous review. Below is a list of comments:
We apologise that you did not receive our full response to your reviewer feedback. Please see our response below.
There is still no line numbering and the diagram presented is not clear. Also, Table 2 has become unreadable
We are agree that the setting out of table one and table two need re configuring. It is not in the form that was submitted. We think that 'sex', 'country' etc are subheadings and the associated categories e.g. male female etc. should then be indented. In other words, presented in the same way that psychological history appears in table 1. Table two appears to have suffered because from indenting issues and does not appear as it was submitted. No doubt this will be attended to in the final copy version however we have alerted the editorial team to the issue. We assume there is no line numbering as that system is not used in this journal.
As for the diagram, this has been re-formatted. We think that this is more clear for the reader. Can you please specify where improvements are required?
The authors also did not answer, on what basis the number of sessions was 12? Why is this? Have any preliminary studies been conducted? Could it be indicated by the literature data?
We apologise that you did not receive our response to your feedback. Our original response: 12 sessions had been used in a previous RCT that the authors had been involved in and found to lead to clinically meaningful results for chronic PTSD. Also the median number of sessions in studies on Ch-PTSD is 18 but they are typically one hour. So the treatment time in the IREM (12- 90 minute sessions) was similar to previous studies (see Ehring et al [1] for list of studies and session information).
"KBdH, informed by the literature, and reviewed by authors HC and CL" - Researchers' abbreviations should be introduced into the publication like all other abbreviations, ie they should be expanded on first use.
We apologise that you did not receive our response to your feedback. While we agree that general practice is to only use abbreviations after they have been introduced in full, we have followed the standard formatting for qualitative papers used in the journal.
Are there references to studies performed [28] on studies conducted in the same population? It is not explained where the health information came from (end they go on to make further gains which was evidenced in the 80% recovery from PTSD at one year follow up) since the interview was conducted 8 weeks beyond the end of the session. The authors also did not respond to this comment.
We apologise that you did not receive our response to your feedback. The reference [28] relates to the complete IREM RCT results. These results include all of the assessment points including up to the 1-year post-treatment assessment. In the second paragraph of the discussion we provide a brief summary of the IREM results including “68% of participants no longer met the criteria at the first follow-up point, and this improved to 81% at the 1-year follow-up assessment.” This data supports the statement that participants made further gains following the end of treatment.
While no patients specifically commented that they felt they needed less sessions, there were 18 participants of the 155 treated who finished treatment early. - why does the number 155 appear?
Thank you for pointing this out. This sentence has been revised to make clear to the reader. We have changed the sentence to “While no patients in this qualitative sample specifically commented that they felt they needed less sessions, in the entire sample of patients treated in IREM (n=155), 18 participants finished treatment early.”
- Ehring T, Welboren R, Morina N, Wicherts JM, Freitag J, Emmelkamp PM. Meta-analysis of psychological treatments for posttraumatic stress disorder in adult survivors of childhood abuse. Clin Psychol Rev. 2014;34(8):645-657.
Reviewer 2 Report
Thank you for your attention to my recommendations. I have no further suggestions.
Author Response
Dear Reviewer 2,
Thank you for your feedback. As you have indicated there are no further revisions required.
Kind regards
Dr Katrina Boterhoven de Haan
This manuscript is a resubmission of an earlier submission. The following is a list of the peer review reports and author responses from that submission.
Round 1
Reviewer 1 Report
Overall Comments: The authors discuss the importance of understanding both patient and therapist views towards initiating and adhering to trauma-focused treatments for patients with sustained PTSD from adverse childhood experiences. This study is novel in that is combines both perspectives to better understand the barriers to engaging in such treatments. The authors provide a detailed account of the robust analysis procedure, adhering to COREQ guidelines. They tie in their findings with other trauma-focused treatment studies. Overall, this manuscript is a unique original contribution to the literature.
Specific Comments:
Abstract:
Last sentence: I am unfamiliar with the use of ‘stabilisation’ in this context; suggest that you clarify. Do you mean patient stabilization?
Introduction:
First para., first sentence: I suggest you be specific to ‘adverse childhood experiences’ (ACEs) and consider adding a statement defining what the term means. I also suggest a statement in regards to prevalence of ACEs.
Methods:
2.1 Study Context and Design
Para 1, first sentence: I imagine IREM is an acronym for the study. Please spell out for first mention.
Para 2: How exactly were participants screened for eligibility and exclusion criteria?
2.2 Participants, first sentence: Add ‘who’ between ‘patients’ and ‘accepted’.
2.3 Semi-Structured Interview: I find this section lacking in detail. For example, I would imagine that therapists are provided different questions than the patients. I would suggest providing a table listing questions that are unique to each and those that are similar.
Results:
First para: I suggest the authors make clear, when they first introduce, which super-ordinate themes are shared by patients and therapist and which one is unique to therapists. Also, I don’t see a need to separate out in your illustrative figure (make one figure vs two). When you combine, you can introduce brackets indicating which 3 super-ordinate themes are shared and which one is therapist only.
Try to strive for equal representation of therapists and patients for the shared subordinate theme quotes.
3.1.2 Therapeutic relationship: Since this is a shared subordinate theme, I expected there to be representative quotes from participants.
3.2.1 Starting Trauma Work, 2nd para.: lines 6 & 7 are italicized.
3.4 Trauma in Context: As is a subordinate theme of ‘Nature of Change,’ make 3.3.1.
3.5 The Changed Self: As is a subordinate theme of ‘Nature of Change,’ make 3.3.2.
3.6 Optimizing the Therapist Role: Make 3.4. Also, make your subsequent subordinate themes structured according to your 3.4.xx formatting.
General: Since you included patients from 3 countries, did you do any comparison sub-analyses to identify any differences?
Discussion:
Other study references: The authors appropriately compare with other studies of trauma treatment approaches. Are any of these specific to Ch-PTSD patients? If so, please indicate.
1st para., last sentence: Change ‘adherence’ to ‘adhering.’
2nd para., first sentence: Add ‘for Ch-PTSD.’
2nd para., 3rd sentence: Change ‘reduction’ to ‘reductions.’
3rd para., 3rd sentence: replace ‘for’ with ‘to engage in.’
3rd para., When mentioning ‘with a stabilization phase’ or ‘phase-based approach’, many readers (including me) may not understand what that means. Can you briefly explain?
Last para. (prior to ‘study limitations’): This paragraph reads to me as being more fitting for the Introduction section.
Limitations, 2nd paragraph: Do you want to counter the ‘lost in translation’ with your participant checking procedures?
Conclusion, 2nd para., 1st sentence: add ‘for Ch-PTSD’ after the word treatments.
General:
Tables: Left columns would read more easily with a left-indentation. Would suggest moving Tables 1 and 2 to Results section.
Reviewer 2 Report
Returning to difficult childhood experiences will always cause anxiety. Carrying out such a process is, however, extremely important from the point of view of the mental health and further functioning of the injured person. Therefore, the conducted research can significantly facilitate this process. However, the presented work has some shortcomings. Below is a list of comments.
There are no verse numbers
The presented diagrams are not very clear
The description of the method lacks information on how the interviews were conducted? What questions were asked? Were questions asked at all? Additionally, some comments from therapists or patients could be clearer if the reader learned the type of questions asked, or at least the way of conducting the conversation. However, in the description of the method, the reader often learns how the responses of patients and therapists were interpreted and checked, which is definitely less interesting.
On what basis was it determined that the number of sessions would be 12?
The description of the method lacks at least residual information on the research carried out earlier and their results. Hence, a lot of information may be incomprehensible if the reader does not have access to the two previous publications referenced by the authors.
Moreover, the references to the studies performed [26] do not coincide with the information indicated in this manuscript. It is not explained where the health information came from (end they go on to make further gains which was evidenced in the 80% recovery from PTSD at one year follow up) since the interview was conducted 8 weeks beyond the end of the session.
"KBdH, informed by the literature, and reviewed by authors HC and CL" - Researchers' abbreviations should be introduced into the publication like all other abbreviations, ie they should be expanded on first use.
While no patients specifically commented that they felt they needed less sessions, there were 18 participants of the 155 treated who finished treatment early. - why does the number 155 appear?
I assume that the purpose of this publication is to define how to improve the method to make it as effective as possible. However, the summary lacks clear information on what should be changed in practice to make this method more effective. The more so because the answers of both groups contain contradictory information.
Reviewer 3 Report
The authors studied the patients’ and therapists’ experiences with trauma-focused treatments in patients with posttraumatic stress disorder from childhood trauma. The subject as well as the whole conception of the experiment seems really interesting and reasonable. However, the issue raised by the authors does not fit within the scope of the journal and the whole manuscript does not provide sufficient scientific soundness which justifies its publication in JCM in present form.
Some suggestions:
1. Introduction:
please verify whether colon is necessary
"i.e." before 23 22 citations is unnecessary, in my opinion
Ethics approval for this study was obtained from the relevant ethics committee in each country. - Please, add the ID of protocols
Eligible participants had a primary diagnosis of PTSD related to trauma experienced before 16 years of age. Please, mention diagnosis criteria which were used.
Table 1 is not visible – reading of the information is difficult, examined factors should be bolded – f.e. gender, relationship status
In the characteristic of sample information about the country should be added
Figure 1, Figure 2: The font could be bigger because the text is hard to read